# Osteopathy in Complex Lymphatic Anomalies

**DOI:** 10.3390/ijms23158258

**Published:** 2022-07-26

**Authors:** Ernesto Solorzano, Andrew L. Alejo, Hope C. Ball, Joseph Magoline, Yusuf Khalil, Michael Kelly, Fayez F. Safadi

**Affiliations:** 1Department of Anatomy and Neurobiology, Northeast Ohio Medical University (NEOMED), Rootstown, OH 44272, USA; esolorzanozepeda@neomed.edu (E.S.); aalejo@neomed.edu (A.L.A.); hball@neomed.edu (H.C.B.); joemagoline@gmail.com (J.M.); ykhalil@neomed.edu (Y.K.); mkelly4@neomed.edu (M.K.); 2Musculoskeletal Research Group, Northeast Ohio Medical University (NEOMED), Rootstown, OH 44272, USA; 3Department of Pediatric Hematology Oncology and Blood, Cleveland Clinic, Cleveland, OH 44195, USA; 4Rebecca D. Considine Research Institute, Akron Children’s Hospital, Akron, OH 44308, USA; 5School of Biomedical Sciences, Kent State University, Kent, OH 44243, USA

**Keywords:** lymphatic, bone, lymphangiogenesis, osteolysis, anomaly

## Abstract

Complex Lymphatic Anomalies (CLA) are lymphatic malformations with idiopathic bone and soft tissue involvement. The extent of the abnormal lymphatic presentation and boney invasion varies between subtypes of CLA. The etiology of these diseases has proven to be extremely elusive due to their rarity and irregular progression. In this review, we compiled literature on each of the four primary CLA subtypes and discuss their clinical presentation, lymphatic invasion, osseous profile, and regulatory pathways associated with abnormal bone loss caused by the lymphatic invasion. We highlight key proliferation and differentiation pathways shared between lymphatics and bone and how these systems may interact with each other to stimulate lymphangiogenesis and cause bone loss.

## 1. Introduction

Complex Lymphatic Anomalies (CLA) are characterized by idiopathic boney lesions caused by abnormal lymphatic invasion. The lymphatic network is a one-way system mainly responsible for maintaining fluid homeostasis, lipid absorption, and immune surveillance [1,2,3]. The lymphatic system absorbs interstitial fluid, a key component of lymphatic fluid (LF), accumulated from the extracellular spaces, which contain various proteins, small molecules, and lipids [4]. LF is transported unidirectionally back into the venous circulation. This network of vessels houses the immune system, which is responsible for foreign body detection and defense. The lymphatic network is present in most tissues except for bone, skeletal muscle, bone marrow, cartilage, and majority of the eye [4,5].

Lymphatic cells originate primarily from the endothelial lining of venous walls during fetal development, yet they are also known to differentiate from non-venous origins such as mesenchymal stem cells and angioblasts [6,7,8,9,10]. Prospero Homeobox 1 (Prox1) is a key transcription factor known to initiate LEC differentiation from blood endothelial cells [11]. After budding from veins, Vascular Endothelial Growth Factor-C (VEGF-C) is the central protein known to drive proliferation and migration of Lymphatic Endothelial Cells (LEC) during embryogenesis [12,13,14]. VEGF-C binding to Vascular Endothelial Growth Factor Receptor-3 (VEGFR-3) activates Phosphatidylinositol-4,5-Bisphosphate 3-Kinase (PIK3) and mammalian Target of Rapamycin (mTOR) pathways that increase Prox1 expression which maintains the lymphatic phenotype. Vascular Endothelial Growth Factor-D (VEGF-D) is also capable of binding to VEGFR-3 to induce lymphangiogenesis, yet VEGF-D-deficient mice can still develop normal lymphatics [15]. When the VEGF-C/VEGFR-3 and Prox1 pathways are blocked, lymphatic development is prevented [16]. 

Stimulation of the receptor activator nuclear kappa beta (RANK) through RANK ligand (RANK-L) is another indispensable signaling mechanism during lymphatic development. RANK-L secretion from hematopoietic lymphoid tissue inducing (LTi) cells and the mesenchymal lymphoid tissue organizer (LTo) cells is crucial for lymph node organogenesis. In response to RANK-L, LEC are known to attract macrophages during development [7]. Dysfunction in the RANK/RANK-L signaling pathway is known to lead to abnormal lymphatic development [17].

In CLAs, abnormal organization and growth of lymphatics occurs in surrounding tissues, including bone. The skeletal system provides structural support for the body in vertebrates while also serving as a mineral storage source [18]. While the periosteum (connective tissue surrounding bone) houses an extensive network of blood and lymphatic vessels, lymphatic protrusion into cortical and trabecular bone does not normally occur. There is a continuous turnover of bone that is maintained through a balance in cytokines regulating crosstalk between bone cells and balancing bone resorption and bone formation. Skeletal remodeling is the result of the tightly regulated coordination of osteoblasts (OBs) that secrete the bone matrix, osteocytes that sense mechanical stressors, and osteoclasts (OCs) that resorb the calcified bone matrix and shape the bone [19]. Disruption of this balance leads to accelerated bone resorption or decreased bone formation throughout the bone [20,21,22]. In rare cases, we can also observe non-generalized osteolysis only affecting specific bone regions, as seen in Complex Lymphatic Anomalies (CLA) [23,24]. 

Activation of transcription factors Runx-2 (Runt-related transcription factor-2) and osterix is essential for OB differentiation [25,26]. OB secrete the bone matrix, which is mainly composed of type 1 collagen and hydroxyapatite [27,28]. In vitro, OB require exposure to β-Glycerophosphate as well as ascorbic acid to grow and differentiate [29,30]. OB differentiation may also be stimulated through activation of the VEGFR-3, yet their auto stimulation through VEGF-C is unlikely [31,32]. After continuous matrix secretion, OB become embedded within the bone and are then differentiated into osteocytes [33]. Osteocytes are key regulators of OB and OC through secretion of RANK-L, osteopontin, macrophage colony-stimulating factor (M-CSF), and sclerostin in response to various stimuli [34,35,36].

Hematopoietic stem cells (HSCs) differentiate into macrophages (pre-osteoclasts) when exposed to M-CSF secreted by OB and/or osteocytes [37,38,39]. These macrophages are further differentiated into mature osteoclasts by exposure to RANK-L, secreted by OB and/or osteocytes, to become mature and active OCs [37]. Inflammatory cytokines such as Interleukin-1 beta (IL-1β), Interleukin-6 (IL-6), and Tumor Necrosis Factor-alpha (TNF-α) are known to stimulate OC maturation while inhibiting OB differentiation and function [18,40]. OC are also capable of auto stimulation through VEGF-C. In OC, RANK-L stimulation triggers secretion of VEGF-C and subsequent activation of its main receptor VEGFR-3, leading to accelerated differentiation and function [41].

While bone and lymphatic cells are regulated by similar cytokines, they do not come in contact with each other in non-pathological conditions. Complex Lymphatic Anomalies (CLA) are characterized by abnormal lymphangiogenic vessels invading bone, resulting in loss of bone through poorly defined mechanisms. The purpose of this review is to present the unique bone phenotype in each CLA and present a new perspective on how the interaction between lymphatics and bone trigger osteolysis.

## 2. Complex Lymphatic Anomalies (CLAs)

CLAs include four diseases with overlapping and distinct clinical features: Gorham-Stout Disease (GSD), General Lymphatic Anomaly (GLA), Kaposiform Lymphangiomatosis (KLA), and Central Conducting Lymphatic Anomaly (CCLA) [42]. The etiology of these diseases has been proven to be extremely elusive over the years due to their rarity and irregular progression. CLAs are believed to be developmental in nature, mainly affecting individuals under the age of 40 and with no known predilection for sex [43,44,45]. Each CLA has a unique lymphatic phenotype, as reported by the International Society for the Study of Vascular Anomalies (ISSVA) [46,47]. CLAs have been associated with somatic activating mutations in genes involved in the Phosphatidylinositol-4,5-bisphosphate 3-kinase, catalytic subunit alpha, and Protein kinase B (PIK3CA-Akt) and Rat sarcoma virus and Mitogen-activated kinase (RAS-MEK) pathways (Table 1) [48,49]. A more complete understanding of the pathology of CLAs will help contribute to the development of new diagnostics and evidence-based therapies to treat soft tissue and boney complications in successfully identified patients. Here, we compiled the current knowledge on each CLA regarding their clinical presentation, lymphatic, and bone phenotype, followed by presentation of possible regulators during bone loss.

During normal lymphatic formation, VEGF-C binds to VEGFR-3 and activates both PIK3CA-AKT and RAS-MEK signaling pathways with a net increase in LEC proliferation, survival, and migration. These pathways are closely linked to multiple lymphangiomas, including CLA, as recently reviewed by Liu et al. [50]. The first somatic activating mutation in PIK3CA was first reported in 2019 on a cohort of GLA patients [49,51]. Since then, multiple activating somatic mutations were also identified in the RAS-MEK pathway [52,53,54,55,56]. These findings were corroborated through selective overexpression of PIK3CA and RAS genes in lymphatic cells of mice and fish models [49,53,57,58]. Since these lymphatic mutations are commonly found in cancer patients, patient biopsy samples can be quickly assessed for activating mutations through cancer diagnostic panels [42]. To date, at least one mutation has been identified in each of the CLA (Table 1).

**Table 1 ijms-23-08258-t001:** Genetic mutations associated with Complex Lymphatic Anomalies. Each CLA has been identified with at least one somatic mutation. GSD mutation: KRAS (Kirsten rat sarcoma virus). GLA mutations: PIK3CA (Phosphatidylinositol-4,5-bisphosphate 3-kinase, catalytic subunit alpha), NRAS (rat sarcoma viral oncogene homolog). KLA mutations: NRAS (rat sarcoma viral oncogene homolog), BAD (BCL2 Associated Agonist of Cell Death), TSC1 (Tuberous sclerosis complex 1), CBL (Cbl Proto-Oncogene). CCLA mutations: EPHB4 (Ephrin type-B receptor 4), ARAF (A-Raf proto-oncogene), MDFIC (MyoD Family Inhibitor Domain Containing).

Type of CLA	Bone Phenotype	Defining Pathology	Somatic Mutations	References
Gorham Stout Disease (GSD)	Localized cortical and trabecular osteolysis	Massive bone destruction and resorption	KRAS	[53,59]
Generalized Lymphatic Anomaly (GLA)	Generalized trabecular osteolysis	Multi-organ, multicentric, proliferative lesions	PIK3CA, NRAS	[49,54,60]
Kaposiform Lymphangiomatosis (KLA)	Generalized trabecular osteolysis	Multi-organ, multicentric, proliferative lesions	NRAS, BAD, TSC1, CBL	[52,55,61,62]
Central Conducting Lymphatic Anomaly (CCLA)	Channel-like osseous lesions	Dilated central conducting channels	EPHB4, ARAF, MDFIC	[56,58,63]

### 2.1. Gorham Stout Disease (GSD)

GSD, also known as “vanishing bone disease”, is an extremely rare condition characterized by idiopathic and progressive osteolysis affecting one or more bones [64,65]. The defining characteristic of GSD is the presence of both cortical and trabecular bone loss caused by progressive invasion of lymphatics into bone (Figure 1) [66]. While other CLAs only develop trabecular bone loss, GSD patients present both cortical and trabecular osteolysis. GSD typically involves one or few bones in a regional distribution. Functional consequences of GSD are determined by the anatomical location and progression of the disease. Major symptoms of GSD patients include bone pain, inflammation, limited movement, skeletal deformity, and chylothorax [67,68]. Chylothorax (accumulation of lymphatic fluid in the thoracic cavity) is a potentially life-threatening complication shared by all CLAs. Skull involvement and fractures are more prevalent in GSD compared with other CLAs [67,69]. Affected bone structures are filled with infiltrative soft tissue significantly more in GSD than other CLAs [66,68]. Chylothorax and spinal instability caused by progressive osteolysis is linked to 33–55% of GSD patient deaths [70]. The mortality of this condition has been shown to be around 13.3%, which indicates a relatively good prognosis due to spontaneous arrest of disease progression [71]. However, the prognosis becomes worse if the spine or thorax is involved due to the risk of neurological complications or chylothorax [71,72].

Recently, a gain-of-function somatic mutation in KRAS (p.G12V) of GSD LECs has been shown to cause lymphatic malformations through hyperactivation of the RAS-MEK pathway [53,59,73]. This pathway is commonly implicated in abnormal lymphangiogenesis and several cancer types, including breast, colorectal, and liver [74]. Most recently, a mouse model with a similar activating somatic mutation in KRAS (p.G12D) was found to mimic the abnormal lymphatics and bone invasion identified in GSD patients [53,59]. Despite the recent advancements, further work is necessary to elucidate how this interaction between lymphatics and bone leads to progressive lymphatic growth and osteolysis.

### 2.2. General Lymphatic Anomaly (GLA)

GLA is characterized by diffuse lymphatic invasion with dilated lymphatic vessels that affect a variety of soft tissues and bone [54,66]. In contrast to GSD, GLA has a generalized distribution with multiorgan involvement containing lytic trabecular boney lesions (Figure 2). The axial skeleton is commonly affected in GLA with a focus on the ribs and spine. This variable presentation affecting multiple sites in the body makes it hard to diagnose due to the variety of symptoms varying from case to case. GLA can occur at any age, often presenting with the highest in children and teenagers with symptoms presenting before the age of 20 [68,75]. Thoracic involvement is a common and serious complication in patients diagnosed with GLA. A poorer prognosis is observed when there is a pleural effusion into soft tissues such as the lungs, heart, or peritoneum [68,76,77]. 

### 2.3. Kaposiform Lymphangiomatosis (KLA)

KLA is a particularly aggressive subtype of GLA that uniquely presents with spindle endothelial cells throughout the malformed lymphatic channels [66,78]. KLA displays a progressive involvement with severe morbidity and a high mortality rate. In a retrospective study of 20 patients between 1995 and 2011, the 5-year survival rate was reported to be 51%, with an overall survival rate of 34%; the cause of death in most instances is due to cardiorespiratory failure caused by consumptive coagulopathy [78]. This highly aggressive subtype presents at a young age, and many complications can arise, including organ failure, pleural and pericardial effusions, ascites, pain, and boney osteolysis [77]. 

The bones affected in KLA reside mainly in the thoracic cavity, and the bone involvement is similar to GLA (Figure 2). While KLA and GLA patients present with similar boney phenotypes, a defining feature that KLA displays its ability to affect multiple organs with pleural effusions, ultimately leading to respiratory distress in the thoracic cavity [68]. KLA diagnosis exhibits lymphatic channels with focal areas of spindle-shaped “kaposiform” endothelial cells, positive for Podoplanin and PROX-1 (markers for lymphatic tissue) [77]. KLA exhibits many features of GLA, such as nonprogressive lesions in bone’s medullary cavity, widespread lymphangiomatosis, and multifocal lymphatic malformations that involve the bones, viscera, thoracic, and abdominal cavities [42,78]. 

KLA is caused by mutations in the RAS viral oncogene homolog (NRAS) and Cbl Proto-oncogene (CBL) genes identified in patient tissue biopsies [49,57,60,77,79]. Because tissue biopsy may involve surgical intervention near vital organs that could lead to significant complications worsening the disease, the use of noninvasive diagnostics for KLA is warranted. Elevated Angiopoietin-2 (Ang-2) (a protein involved in blood vessel maturation and stability) was found to serve as a reliable biomarker for KLA [80,81]. In addition, Ang-2 levels normalize upon successful treatment suggesting its potential to track disease progression [82]. Interestingly, Ang-2 has been reported to have a negative effect on vessel maturation and stability [83]. Despite these discoveries, further research is necessary to determine Ang-2′s involvement in KLA pathogenesis.

### 2.4. Central Conducting Lymphatic Anomaly (CCLA)

CCLA, also known as channel-type LM, is a rare subset of disorders that are caused by dysfunction of the thoracic duct and/or cisternae chylae. These result in reflux and leakage of lymphatic fluid around the lungs, heart, abdomen, and legs, often with lymphedema involving bilateral lower extremities [84]. CCLA’s distinguishing clinical feature consists of dilated central conducting lymphatic channels that lead to improper fluid transport (Figure 3). Common complications include chylothorax, pulmonary lymphangiectasia, chylous ascites, protein-losing enteropathy, lymphedema, cutaneous vesicles, or superficial chylous leaks [85]. Boney lesions have also been reported in patient vertebrae [84]. These osseous changes include focal areas of hyperlucency due to dilated intraosseous lymphatic channels with a more permeative appearance (unpublished observations and personal communication with Dr. Michael Kelly). CCLA can occur at any age; however, the majority of cases are reported in pediatric patients below the age of 20. CCLA has been recently classified as a CLA due to multiorgan invasive profile, yet it is commonly referred to as channel-type lymphatic malformation.

## 3. Lymphatic Bone Invasion

Somatic activating mutations in the PIK3CA or RAS/MEK pathways have been determined to cause abnormal lymphatic phenotype as seen in CLAs [49,51,54,56,60,61,86,87,88]. In order to replicate the phenotype caused by activating PIK3CA mutations in patients, a mouse model overexpressing this mutation in cells expressing Prox1 (LEC marker) was generated. This model selectively expressed the mutation in lymphatic tissue leading to lymphatic hyperplasia and dysfunction similar to that observed in GLA [49]. Mice treated with the mTOR inhibitor Sirolimus (Rapamycin), a pharmaceutical therapy used with success in some patients with CLAs [68,69,70,71,72,73,84], proved to both prevent and attenuate lymphatic anomalies [49]. In addition, Mitogen-Activated Protein Kinase (MEK) inhibitors have also been successful in treating RAS-MEK mutations in CLA patients, CCLA zebrafish model, and GSD mouse model [53,55,56,58]. For this reason, CLA association with activating somatic mutations (particularly in the PIK3CA-Akt and RAS-MEK pathways) is now unquestionable. It is currently hypothesized that these patient mutations take place in peripheral lymphatics surrounding affected tissue, but further research is necessary to validate this hypothesis [89]. 

In order to assess lymphatic invasion into bone, mice overexpressing VEGF-C in osterix-expressing cells (early OB marker [90]) were found to stimulate lymphatic invasion from peripheral lymphatics past muscle, cortical bone, and into the bone medullary cavity [89,91]. This model presented the ability of peripheral lymphatics to proliferate and invade bone. While this model may not directly represent a CLA, it supports the theory that aberrant lymphatics grow from pre-existing lymphatic vessels. In addition, this study further supports the fact that the sole presence of LEC in bone (with/without mutation) leads to aberrant bone loss, while the migration into bone may be triggered by somatic mutations. 

Another possible explanation to determine the presence of lymphatics in bone could be due to abnormal LEC migration during development. This hypothesis could explain the reason why there are CLA subtypes that only affect trabecular bone (GLA/KLA) while also justifying the abnormal inherent characteristics found in bone cells isolated from GSD patients [54,67,68,92,93,94]. Further research focused on understanding the source of invading lymphatics might facilitate patient diagnosis and development of new treatments.

## 4. Current Knowledge of CLA Osteopathy

Despite the latest discoveries of somatic mutations in each disease, the bone loss mechanism seen in CLA remains to be elucidated. To date, bone loss identified in CLA has not been linked to a somatic genetic mutation in bone cells. TNF Receptor Superfamily Member 11a (TNFRS11) and Triggering Receptor Expressed in Myeloid cells 2 (TREM2) mutations were found in a GSD patient biopsy sample. These mutations are known to lead to bone loss through increased OC differentiation [95,96] and excessive osteolysis [97]. Since these mutations were only reported in one GSD patient, we cannot determine that these mutations are associated with CLAs or which cell type is involved.

Prior to the presentation of abnormal lymphatics, there is no abnormal skeletal phenotype. This leads us to believe that the direct contact between lymphatics and bone is the trigger of bone loss. In order to explore the effect lymphatic invasion has on bone, non-pathogenic lymphatic cell lines were intra-tibially injected in mice [98,99]. This approach procured massive osteolysis in mice which led to complete bone loss within weeks. Additionally, this effect was closely linked to OC activation through M-CSF secretion from LEC [98]. This study proved that interactions between lymphatics and bone will lead to bone loss without the need of a pre-existing somatic mutation in LEC. While M-CSF may play a key role in lymphatics inducing bone loss, this alone cannot explain the phenotype seen in CLAs.

Increased OC activity is reported in various CLAs [7,40,89,100] while also reporting little to no bone formation to replace lost bone in GSD [101,102]. There are few reports that successfully isolated bone progenitor cells from CLA patients and differentiated them into OC or OB in culture. Marked osteoclastic resorption in both trabecular and cortical bone was initially reported in GSD patient biopsy [45]. This group performed six separate biopsies per patient through histological analysis, and all were consistent with massive osteoclast-mediated bone resorption [45,103]. Cultured peripheral mononuclear cells (PMCs) from GSD patients compared to healthy controls boosted OC differentiation and function [40]. A recent study repeated this isolation approach in multiple patients and obtained similar results [67]. Both studies mention structural differences between patient and control OC. OC from these patients appeared more spindle shaped, suggesting that these cells have a higher motility rate which increases resorption activity by OC. Microarray analysis of GSD OC revealed dysregulated genes caused by the disease. Among these, low-density lipoprotein receptor-related protein 6 (LRP6), an important component in OC maturation [104], was upregulated in GSD OC [67]. 

Even though OC are often overactive in GSD, this is not a universal finding in all GSD patients studied [42]. This inconsistency can possibly be attributed to samples from patients with varying disease activity/progression. Patients with accelerated bone loss would be expected to have high OC activity. Despite inconsistent reports, overactive OC are believed to be the main drivers of the osteolysis mechanism observed in CLA patients [89]. Delayed fracture healing can be attributed to reduced OB mineral deposition, as reported in GSD patients [67,94]. Recently, Rossi et al. successfully isolated PMCs and Mesenchymal Stem Cells (MSC) from GSD patients and differentiated them into OC and OB, respectively [67,94]. GSD OC demonstrated enhanced differentiation, while GSD OB had lower mineral deposition and increased RANK-L secretion in comparison to healthy controls [67]. Furthermore, an epigenetic assessment of cells from these patients identified micro-RNA (miR) dysregulations. To date, various miRs are known to regulate various aspects of OB and OC differentiation and function [105]. When assessing miRs from GSD bone cell cultures, Rossi et al. identified various miRs capable of regulating OC formation and activity (miR-1246, miR-1-3p, and miR-137-3p) and OB differentiation and function (miR-204a-5p, miR-615-3p, and miR-378a-3p) [94]. Together, these data suggest GSD patients have inherently dysregulated bone progenitor cells. These changes could be influenced by dysregulated lymphatic secretion of cytokines normally involved in bone homeostasis (such as VEGF-C or RANK-L), as further explored in this review.

## 5. Proposed CLA Osteopathy

Under normal conditions, lymphatic and bone cells do not interact, but both bone and lymphatics readily secrete various cytokines and growth factors that trigger lymphangiogenesis and bone resorption (Figure 4). Since normal lymphatics can induce abnormal bone loss [89,91,98], we hypothesize that lymphatic invasion towards bone is mainly attributed to somatic mutations. After reaching the bone, direct interaction between lymphatics and bone cells will result in various forms of osteolysis, as seen in CLAs.

Lymphangiogenesis is highly driven by exposure to VEGF-C. OC secrete VEGF-C when stimulated with RANK-L in an SRC Proto-Oncogene, Non-Receptor Tyrosine Kinase (Src) phosphorylation-dependent manner [41,106]. Additionally, VEGF-C treatment is known to stimulate OC in a similar manner as M-CSF [107]. Macrophages (OC progenitors) and MSC are also known to secrete various VEGF factors, including VEGF-C [8,108]. These VEGF-C sources might readily further stimulate lymphatic invasion and trigger bone resorption.

RANK-L and M-CSF stimulation of both lymphatic and OC may play a critical role during lymphatic invasion and osteolysis. Within the lymphatic system, RANK/RANK-L stimulation is necessary for lymphatic node development [7]. Mice deficient for RANK-L or its receptor failed to develop lymph nodes, illustrating the necessity of RANK-L for both OC differentiation and function and lymphatic development [17,109,110]. RANK-L stimulation of LEC induces local macrophage aggregation and could facilitate osteoclast fusion [7]. Similar to OB and osteocytes, LEC secrete M-CSF and RANK-L in vitro [41,98,111]. Subsequently, lymphatic vessels transport immune cells such as helper T-cells, which secrete RANK-L and IL-6 (stimulants of OC differentiation) [112]. Upon facilitated exposure during CLA, OC would be activated by M-CSF, RANK-L, IL-6, and VEGF-C from lymphatics and trigger aggressive bone resorption. Similarly, lymphangiogenesis would be activated through exogenous VEGF-C stimulation from activated OC. Furthermore, MSC have been reported to develop lymphatic endothelial phenotypes (expression of Prox1 and Lymphatic Vessel Endothelial Hyaluronan Receptor 1(Lyve1)) with enhanced lymphatic regeneration when stimulated with VEGF-C [113,114]. This mechanism could provide an alternative route for the aggregation of lymphatic cells in affected regions. While likely, further studies are needed to confirm this interaction between bone and lymphatics.

An animal model for bone lymphatic invasion overexpressing VEGF-C in osterix expressing cells (OB and osteocytes) developed lymphatics in both cortical and trabecular bone (such as GSD) within 35 days through OC activation [89,91]. In comparison to the previous genetic mutation models, this model induces osteolysis and develops lymphatic invasion similar to GSD, while it does not represent mutations associated with patients. While this model does not directly compare to a CLA, it provides a reliable method to induce peripheral lymphatic invasion into bone while shedding some light on the role of VEGF-C during bone loss caused by lymphatic invasions.

Inflammation within the CLA lesion is a common complication found among patients and may trigger disease activity. Cytokines involved in inflammation, such as TNF-α, IL-1β, and IL-6, are known to stimulate OC differentiation in the affected area and increase bone resorption [115]. PMCs, OC progenitors, isolated from GSD patients showed increased sensitivity to stimulation of TNF-α, IL-1 β, and IL-6 in comparison to PMCs isolated from control subjects [40]. Similarly, inflammatory cytokines are involved in the inhibition of OB differentiation and function, reducing bone matrix deposition and subsequently decreasing bone density and bone formation [115]. In addition, IL-6 is also known to stimulate VEGF-C expression by LEC leading to lymphangiogenesis [116]. Lastly, inflammation is known to attract and dilate lymphatic vessels in various inflammatory diseases [117,118,119,120,121]. Based on our current knowledge, inflammation could also play a role in bone loss during lymphatic invasion in CLAs.

## 6. Osteocyte Osteolysis in GSD

To date, the differences in bone involvement between GSD and other CLAs remain to be delineated. Due to its regionally aggressive cortical and trabecular bone loss, GSD has stirred the greatest interest to elucidate its osteopathy. Assessment of six GSD patients’ serum revealed higher levels of IL-6, Vascular Endothelial Growth Factor-A (VEGF-A), cross-linked telopeptide of type 1 collagen (ICTP), and sclerostin [67]. IL-6 is a cytokine linked to inflammation known to stimulate bone resorption through RANK-L secretion from OB and osteocytes [121,122]. VEGF-A is a well-known inducer of blood vessel angiogenesis and is likely linked to angiomatous proliferation in CLA patients. ICTP is a marker for OC activity, while sclerostin is a protein commonly secreted from osteocytes that binds and inhibits OB [123]. These serum markers found in GSD attribute to overall bone loss in patients and can explain the aggressive osteolytic phenotype. Regional involvement associated with GSD can be attributed to localized lymphatic invasion caused by somatic mutations. Despite this, cortical involvement remains an outstanding feature of GSD that remains to be understood. 

Out of all the CLAs, GSD is the only condition to report elevated sclerostin. Along with OB inhibition, sclerostin osteocyte stimulation is known to induce osteocytic osteolysis, where osteocytes resorb surrounding bone (perilacunar remodeling) [36,124]. Osteocytic osteolysis has been found to decrease bone mineralization with increased bone resorption and reduced bone healing during hyperthyroidism, osteoporosis, tumors, and calcium deficiency [125]. Histological analysis of GSD patients revealed enlarged lacunae in comparison to control patients [67]. Osteocytes are exclusively found embedded within the cortical bone. For this reason, osteocyte lacunar resorption could debilitate the cortical structure, facilitate lymphatic invasion, and stimulate bone loss, as found in GSD. Despite these findings, further research is necessary to delineate the role of osteocytes in CLAs.

## 7. Summary

CLAs are rare diseases associated with lymphatic invasions that trigger bone loss. Each CLA has a unique lymphatic and bone profile that defines the disease. Trabecular bone is almost always affected, while cortical bone loss is exclusively related to GSD. The mechanism of bone loss in CLA has been largely associated with overactivation of OC through unknown means. In this review, we presented possible interactions between lymphatics and bone cells during invasion and bone loss. Based on studies presented in this review, we hypothesize that VEGF-C and RANK-L secretion from both lymphatic and bone cells stimulate OC and lymphatic cells to increase proliferation and function, leading to lymphatic invasion and bone resorption, respectively. In addition, we explored the role that osteocytes may have in facilitating cortical bone loss during GSD.

## Figures and Tables

**Figure 1 ijms-23-08258-f001:**
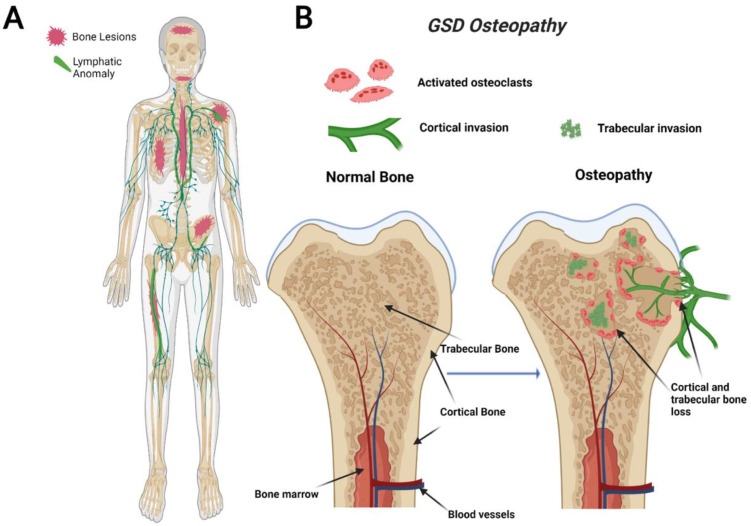
Visual representation of symptoms associated with Gorham-Stout Disease (GSD). GSD patients have regionally aggressive bone loss caused by abnormal lymphatic invasion (**A**). During GSD, both cortical and trabecular bone are resorbed due to lymphatic invasion (**B**).

**Figure 2 ijms-23-08258-f002:**
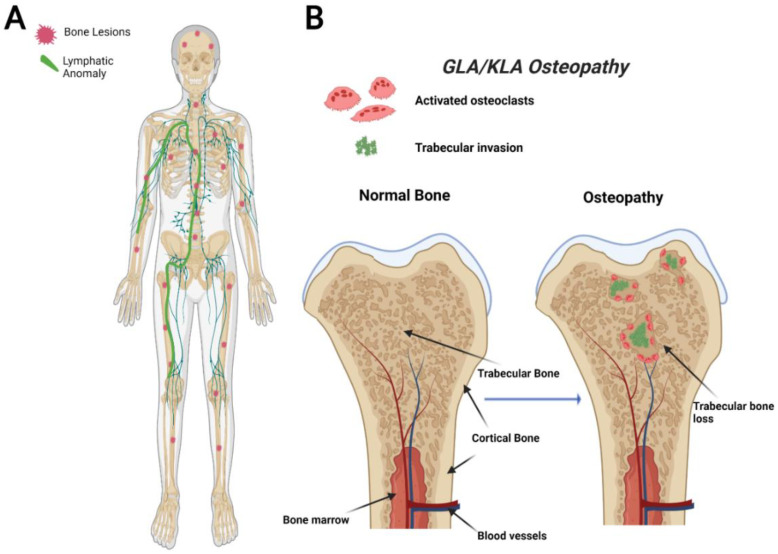
Visual representation of symptoms associated with Generalized Lymphatic Anomaly (GLA) and Kaposiform Lymphangiomatosis (KLA). GLA and KLA patients have generalized trabecular lytic lesions and lymphatic invasion in the thoracic cavity (**A**). Multifocal trabecular invasion is a common phenotype in GLA/KLA patients (**B**).

**Figure 3 ijms-23-08258-f003:**
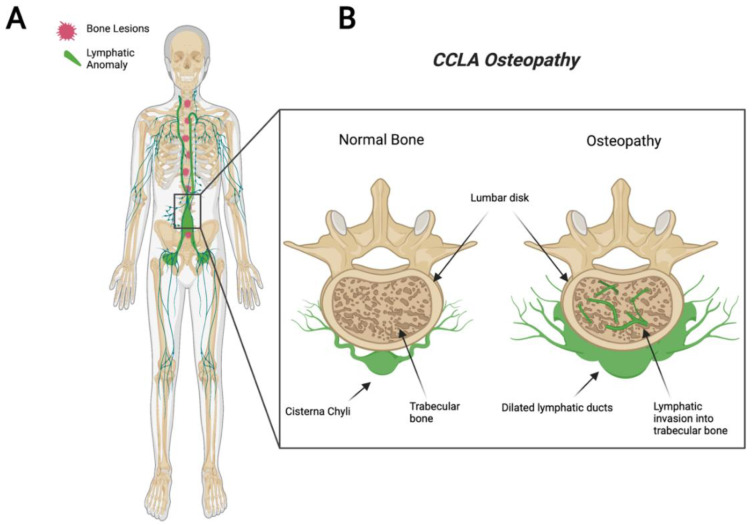
Visual representation of symptoms associated with Central Conducting Lymphatic Anomaly. CCLA patients fail to transport lymphatic fluid and suffer from dilated thoracic ducts. Dilated ducts invade proximal tissue and can cause boney lesions (**A**). Channel-like osseous lesions are found in the trabecular bones (**B**).

**Figure 4 ijms-23-08258-f004:**
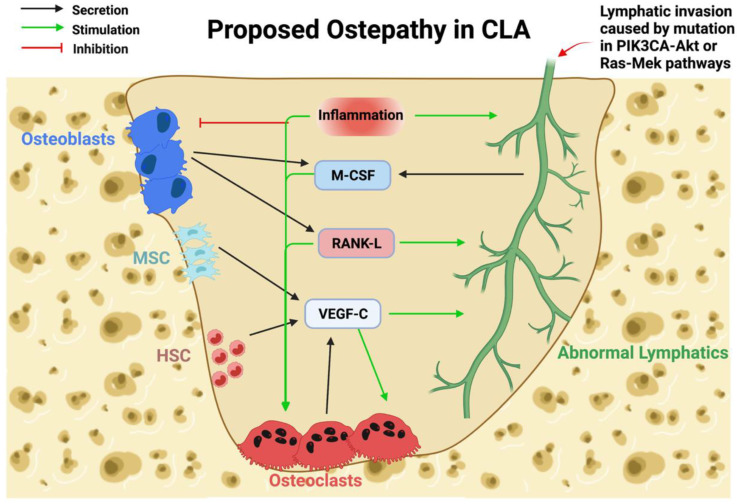
Proposed mechanism of bone loss in CLAs. Cytokines normally secreted by bone cells and lymphatics would stimulate bone resorption during lymphatic invasion. Lymphatic invasion, caused by activating somatic mutations, enables direct contact with bone and triggers resorption. Osteoblasts (OB) secrete M-CSF and RANK-L, which would stimulate Osteoclast (OC) function and lymphatic invasion. Mesenchymal Stem Cells (MSC) and Hematopoietic Stem Cells (HSC) secrete VEGF-C and stimulate OC function and lymphatic invasion. OC secretion of VEGF-C stimulates OC differentiation and lymphatic invasion. Lymphatic secretion of M-CSF stimulates OC differentiation. Inflammation containing various cytokines known to inhibit OB while stimulating OC differentiation and function. Inflammation positively regulates lymphangiogenesis of normal lymphatics.

## Data Availability

Not applicable.

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
