# Peer review of "Osteopathy in Complex Lymphatic Anomalies"

_ijms, 2022, doi:10.3390/ijms23158258_

Round 1

Reviewer 1 Report

This is a comprehensive review on complex lymphatic anomalies (CLA) subtypes – GLA, GSD, KLA and CCLA -which focuses on the different type of osteopathy associated with these diseases. Possible mechanisms for osteopathy and osteolysis in CLA are discussed in great detail. 

The main problem of this review which needs to be addressed is the literature cited. The goal of a review is to serve as a summary of the relevant literature in the field. Citing a review in a review is not a good practice and should be avoided – the goal is to highlight the first report (and additional most significant studies) so that the reader that is new to the field can become familiar with the milestone discoveries. The first 4 pages have a list of references that are not always pertinent or accurate.

Few examples are: the first description of GSD is in Gorham, Stout et al, 1955 – for KLA is in Crouteau at al 2014 – the first report of NRAS mutation in CLA is in Manevitz-Mendelson et al., 2018 (this reference is not included in the table of mutations). Page 3, line 109: refences in 35,36 are not related to the discovery of Pik3CA mutations in GLA patients in 2019..; line110-111 – the references cited (37-40) seem out of place as these studies are not focused on RAS-MEK pathway mutations and some of these references are not on CLAs..

Other minor issues are:

-there is no mention of the ISSVA classification.

-page1 line 31-33 – it is incorrect to say that the lymphatic system is not present in the bone as lymphocytes are part of the lymphatic system and are present in the bone marrow.  I would rephrase to say ‘lymphatic vessel networks are not present in bone.. ‘

-I suggest adding more details on the defects of the lymphatic vessels in CLA and what distinguishes them from other lymphatic anomalies.

-Page 5 line 191: Angiopoietin2 negatively affects vessel maturation and stability as it is involved in vessel proliferation, angiogenesis and permeability – revise text

-it may be worth discussing Angpt2 findings reported in the article: Crane J et al., 2020 PMCID: PMC8554683

Author Response

7/7/2022

Daniel Wu,

Managing Director

Dear Ms. Wu,

We wanted to thank you and the reviewers for your insightful comments on our manuscript entitled “Osteopathy in Complex Lymphatic Anomalies” (IJMS-1751839). The feedback have greatly improved our manuscript.

Please find below our responses to the reviewers’ comments.

Reviewer 1 Comments and Suggestions

General reviewer comments: “This is a comprehensive review on complex lymphatic anomalies (CLA) subtypes – GLA, GSD, KLA and CCLA -which focuses on the different type of osteopathy associated with these diseases. Possible mechanisms for osteopathy and osteolysis in CLA are discussed in great detail.”

Authors’ response:  We appreciate the comments provided by the reviewer. Below we addressed each individual comment and highlighted changes made in red font in the revised manuscript.

Reviewer comment-1: “The main problem of this review which needs to be addressed is the literature cited.”

Authors’ response-1:  We apologize for the confusion regarding the literature cited. Hence, all references were verified for relevance and accuracy.

Reviewer comment-2: “The goal of a review is to serve as a summary of the relevant literature in the field. Citing a review in a review is not a good practice and should be avoided – the goal is to highlight the first report (and additional most significant studies) so that the reader that is new to the field can become familiar with the milestone discoveries. The first 4 pages have a list of references that are not always pertinent or accurate.”

Author’s response-2: Previously cited articles (original manuscript) referenced to 1, 14, 28, and 30 have been replaced with relevant articles (highlighted in red). References such as 29, 51, and 89 were intentionally included since these references are present in one or numerous specific case reports.

Reviewer comment-3: “Few examples are: the first description of GSD is in Gorham, Stout et al, 1955 – for KLA is in Crouteau at al 2014 – the first report of NRAS mutation in CLA is in Manevitz-Mendelson et al., 2018 (this reference is not included in the table of mutations).”

Authors’ response-3: We apologize for missing important references. References mentioned above have been included in the revised manuscript.

Reviewer comment-4: “Page 3, line 109: references in 35,36 are not related to the discovery of Pik3CA mutations in GLA patients in 2019..; line110-111 – the references cited (37-40) seem out of place as these studies are not focused on RAS-MEK pathway mutations and some of these references are not on CLAs..”

Authors’ response-4: Sorry again for the confusion. References were adjusted as recommended.

Reviewer comment-5: “There is no mention of the ISSVA classification”

Authors’ response-5: The ISSVA classification was included in the revised manuscript.

Reviewer comment-6: “page1 line 31-33 – it is incorrect to say that the lymphatic system is not present in the bone as lymphocytes are part of the lymphatic system and are present in the bone marrow.  I would rephrase to say ‘lymphatic vessel networks are not present in bone..”

Authors’ response-6: We agree with the reviewer's comment. Suggested rephrasing was added to the revised version of the manuscript.

Reviewer comment-7–“I suggest adding more details on the defects of the lymphatic vessels in CLA and what distinguishes them from other lymphatic anomalies”

Authors’ response-7: The authors have chosen not to describe the lymphatic phenotype of each CLA in detail since these phenotypes have been recently reviewed [1-4 below] and would divert the focus from the main theme of the review addressing the differences in osteopathy between CLAs.

  1. Makinen, T.; Boon, L. M.; Vikkula, M.; Alitalo, K., Lymphatic Malformations: Genetics, Mechanisms and Therapeutic Strategies. Circ Res 2021, 129, (1), 136-154.
  2. Lee, E.; Biko, D. M.; Sherk, W.; Masch, W. R.; Ladino-Torres, M.; Agarwal, P. P., Understanding Lymphatic Anatomy and Abnormalities at Imaging. Radiographics 2022, 42, (2), 487-505.
  3. Snyder, E. J.; Sarma, A.; Borst, A. J.; Tekes, A., Lymphatic Anomalies in Children: Update on Imaging Diagnosis, Genetics, and Treatment. AJR Am J Roentgenol 2022, 218, (6), 1089-1101.
  4. Gorostidi, F.; Glasson, N.; Salati, V.; Sandu, K., Pediatric vascular anomalies with airway compromise. J Oral Pathol Med 2022.

Reviewer comment-8- “Page 5 line 191: Angiopoietin2 negatively affects vessel maturation and stability as it is involved in vessel proliferation, angiogenesis and permeability – revise text

-it may be worth discussing Angpt2 findings reported in the article: Crane J et al., 2020 PMCID: PMC8554683”

Authors response-8: We agree with the reviewer, more information regarding Angiopoietin2 function is now included in the revised manuscript with the appropriate citation.

In summary, the authors would like to thank the reviewers for their insightful comments. We believe we addressed all comments and concerns adequately. We hope our manuscript is now acceptable for publication in the International Journal of Molecular Sciences.

Sincerely yours,

Fayez Safadi, Ph.D., FASBMR

Reviewer 2 Report

This review sets out to summarise existing literature in the field of complex lymphatic anomalies (CLA) and their relationship with osteopathic lesions. The authors briefly introduce the lymphatic system, how it arises and the key signalling pathways and factors necessary for lymphatic development. They then describe the link between lymphatic vessels and bone in the context of CLA and introduce the concept of overlap in cytokines that regulate the two systems, including a short summary of osteoclast and osteoblast differentiation. The authors outline the four main classes of CLAs and their defining features, including helpful schematics to demonstrate the differences in osteopathy between the classes. Further sections in the review outline current knowledge of the underlying mechanisms of CLA osteopathy, summarising both mouse models and patient data. Finally, they propose a model where osteopathy and aberrant lymphatic growth within bone are mutually enhanced by their shared cytokines.

The authors have done a reasonable job in setting out the review; the structure is logical and comprehensively covers the topic. The nature of rare diseases such as CLA dictates that they are not well characterised, making it difficult to build a comprehensive picture of their aetiology. This review draws together existing studies and focusses on the bone phenotype that presents in each class of CLA. This emphasis differentiates it from a recent review by Makinen et al (Circ Res, 2021; Lymphatic Malformations: Genetics, Mechanisms and Therapeutic Strategies) that also covered the topic of CLAs.

Cited references are generally comprehensive and relevant although some have not been included or are used inappropriately (for example Reference #13 is not really a general reference for the composition of osteoblast secreted bone matrix). With respect  to the citations not included, I would suggest references that attribute the initial discoveries of Prox1, Cbfa1/Runx2 and osterix should be included in the introductory section (e.g., Wigle and Oliver, Cell 1999, Ducy et al, Cell 1997; Komori et al Cell 1997; Nakashima et al Cell 1997 etc.).

In summary, this review summarises existing knowledge and adds to the literature with a model of how lymphatic vessels and bone interact to precipitate osteolysis.

Specific comments:

Line 33: It is now accepted that LEC arise from other sources ie, not exclusively from venous endothelium. The authors should mention these sources or at least acknowledge their existence.

Line 65: This statement needs a reference.

Line 73 and thereafter: Macrophage stimulating factor should be referred to as M-CSF not MCS-F. This should also be corrected in Figure 4.

Line 98: This reference is incorrect and should be reference #39.

Lines 204 &209: References #64 and #69 are duplicated.

Lines 342-344: Are MSCs an alternative source of LEC? If so, perhaps this could be introduced in the introductory paragraph.

Author Response

7/7/2022

Daniel Wu,

Managing Director

Dear Ms. Wu,

We wanted to thank you and the reviewers for your insightful comments on our manuscript entitled “Osteopathy in Complex Lymphatic Anomalies” (IJMS-1751839). The feedback have greatly improved our manuscript.

Please find below our responses to the reviewers’ comments.

Reviewer 2 Comments and Suggestions

General reviewer comments: “This review sets out to summarise existing literature in the field of complex lymphatic anomalies (CLA) and their relationship with osteopathic lesions. The authors briefly introduce the lymphatic system, how it arises and the key signalling pathways and factors necessary for lymphatic development. They then describe the link between lymphatic vessels and bone in the context of CLA and introduce the concept of overlap in cytokines that regulate the two systems, including a short summary of osteoclast and osteoblast differentiation. The authors outline the four main classes of CLAs and their defining features, including helpful schematics to demonstrate the differences in osteopathy between the classes. Further sections in the review outline current knowledge of the underlying mechanisms of CLA osteopathy, summarising both mouse models and patient data. Finally, they propose a model where osteopathy and aberrant lymphatic growth within bone are mutually enhanced by their shared cytokines.”

“The authors have done a reasonable job in setting out the review; the structure is logical and comprehensively covers the topic. The nature of rare diseases such as CLA dictates that they are not well characterised, making it difficult to build a comprehensive picture of their aetiology. This review draws together existing studies and focusses on the bone phenotype that presents in each class of CLA. This emphasis differentiates it from a recent review by Makinen et al (Circ Res, 2021; Lymphatic Malformations: Genetics, Mechanisms and Therapeutic Strategies) that also covered the topic of CLAs.”

Authors’ response: We appreciate the comments provided by the reviewer. Below we addressed each individual comment and highlighted changes made in red font in the revised manuscript.

Reviewer comment-1: “Cited references are generally comprehensive and relevant although some have not been included or are used inappropriately (for example Reference #13 is not really a general reference for the composition of osteoblast secreted bone matrix)”

Authors’ response-1: We apologies about the incorrect citation. Corrected references were provided in the revised manuscript.

Reviewer comment-2: “With respect to the citations not included, I would suggest references that attribute the initial discoveries of Prox1, Cbfa1/Runx2 and osterix should be included in the introductory section (e.g., Wigle and Oliver, Cell 1999, Ducy et al, Cell 1997; Komori et al Cell 1997; Nakashima et al Cell 1997 etc.)”

Authors’ response-2: We thank the reviewer for his/her insightful comment regarding the citation of the initial discoveries of Prox1, Cbfa1/Runx2 and osterix. In the revised manuscript, references were added as recommended.

Specific comments:

Reviewer comment-3: “Line 33: It is now accepted that LEC arise from other sources ie, not exclusively from venous endothelium. The authors should mention these sources or at least acknowledge their existence”

Authors’ response-3: Non-venous origins for LEC were mentioned and cited in the introduction section of the revised manuscript as recommended.

Reviewer comment-4: “Line 65: This statement needs a reference.”

Authors’ response-4: A reference was added as suggested.

Reviewer comment-5: “Line 73 and thereafter: Macrophage stimulating factor should be referred to as M-CSF not MCS-F. This should also be corrected in Figure 4.”

Authors’ response-5: Changes regarding M-CSF were made as recommended.

Reviewer comment-6: “Line 98: This reference is incorrect and should be reference #39.”

Authors’ response-6: Reference was corrected as recommended.

Reviewer comment-7: “Lines 204 &209: References #64 and #69 are duplicated.”

Authors’ response-7: Sorry for the confusion. Duplicate reference was omitted.

Reviewer comment-8: “Lines 342-344: Are MSCs an alternative source of LEC? If so, perhaps this could be introduced in the introductory paragraph.”

Authors’ response-8: MSCs can be an alternative source of LEC. This information was included in the introductory paragraph of the revised manuscript.

In summary, the authors would like to thank the reviewers for their insightful comments. We believe we addressed all comments and concerns adequately. We hope our manuscript is now acceptable for publication in the International Journal of Molecular Sciences.

Sincerely yours,

Fayez Safadi, Ph.D., FASBMR
